# Fast Control for Backlight Power-Saving Algorithm Using Motion Vectors from the Decoded Video Stream

**DOI:** 10.3390/s22197170

**Published:** 2022-09-21

**Authors:** Shih-Lun Chen, Tsung-Yi Chen, Ting-Lan Lin, Chiung-An Chen, Szu-Yin Lin, Yu-Liang Chiang, Kun-Hsien Tung, Wei-Yuan Chiang

**Affiliations:** 1Department of Electronic Engineering, Chung Yuan Christian University, Chung Li City 32023, Taiwan; 2Department of Electronic Engineering, National Taipei University of Technology, Taipei 10608, Taiwan; 3Department of Electronic Engineering, Ming Chi University of Technology, New Taipei City 24301, Taiwan; 4Department of Computer Science and Information Engineering, National Ilan University, Yilan City 26047, Taiwan; 5National Synchrotron Radiation Research Center, Hsinchu City 30076, Taiwan

**Keywords:** LCD (liquid crystal display), motion vectors, backlight power-saving, fast algorithm

## Abstract

Backlight power-saving algorithms can reduce the power consumption of the display by adjusting the frame pixels with optimal clipping points under some tradeoff criteria. However, the computation for the selected clipping points can be complex. In this paper, a novel algorithm is created to reduce the computation time of the state-of-the-art backlight power-saving algorithms. If the current frame is similar to the previous frame, it is unnecessary to execute the backlight power-saving algorithm for the optimal clipping points, and the derived clipping point from the previous frame can be used for the current frame automatically. In this paper, the motion vector information was used as the measurement of the similarity between adjacent frames, where the generation of the motion vector information requires no extra complexity since it is generated to reconstruct the decoded frame pixels before the display. The experiments showed that the proposed work can reduce the running time of the state-of-the-art methods by 25.21% to 64.22%, while the performances are maintained; the differences with the state-of-the-art methods in PSNR are only 0.02~1.91 dB, and those in power are only −0.001~0.008 W.

## 1. Introduction

With the development of technology, portable consumer electronic products not only use a large number of liquid crystal displays (LCDs), but also combine this technology to develop more diversified products, such as wearable devices, smartphones, tablet PC, notebook computers, etc. Moreover, in many application environments of the Internet of Things (IoT), the sensor’s lifetime is critical. One important factor is its power supply [1]. The available resources are limited in the device, and achieving lower power consumption on the device is also important [2]. For all kinds of products, battery life is obviously one of the critical factors when considering whether to buy. Displays are important components of computers, pads, and mobile phones. However, the displays also consume a large portion of the power of those applications. The TFT LCD is a non-self-luminance panel that requires an LED backlight module to display [3]. Compared with traditional LCD and LED technologies, mini-LED backlight technology has become a research hotspot because of its size advantage (100–200 µm), which can be combined with quantum dot technology [4]. However, behind the significant improvement in the display performance of elements such as contrast ratio, color gamut, and dark state, the halo effect and excessive power consumption are the main problems. Therefore, how to reduce the power consumption of the displays and maintain good image quality becomes an important research issue. The technique to tackle the problem is called the backlight dimming algorithm, or backlight power-saving algorithm. Backlight dimming technology can be implemented by either local dimming algorithm or global dimming algorithm. In the local dimming technique, a local dimming backlight system for mini-LEDs has also been proposed [5]. The local dimming algorithm divides the frame into a number of small blocks, then controls the backlight brightness of each block separately. This method reduces the brightness of the backlight locally, to maintain the quality of the image and effectively reduce the power. However, this method induces the large number of complex calculations, which in turn further sacrifices the power. On the other hand, the global dimming algorithm only controls the brightness of the backlight unit for the whole frame. It has lower complexity and is widely used in various LCD systems. Therefore, the global dimming algorithm is an effective way to reduce the LCD backlight power consumption, and the base architecture of the proposed work.

The work in [6] proposed a backlight dimming algorithm that reduced the power by up to 50%. For the improvement of the hardware, the literature [7] combined the characteristics of the switching TFT and the compensation circuit mechanism to realize the drive circuit to reduce the power consumption of the mini-LED display. In [8], the authors realize a highly dynamic mini-LED backlight technology using point spread function (PSF) theory. In [9], a two-dimensional (2D) adaptive dimming technique for an RGB-LED backlight is proposed to achieve a high dynamic contrast ratio. In [10], detection relative ratios capture scene features and patterns, resulting in an extremely high dynamic range. In [11], the video transmission system with autonomously controlled throughput is proposed. The work in [12] utilized current compensation technique for X-Y channels in an LED backlight system with I-V characteristics. In [13], I2GEC (Gray Level Error Control based on Image Integrity) defines the target PSNR. The clipping points are set decreasingly until the resulting PSNR satisfies the target PSNR. In the work of MGEC [14], the frame is divided into 4 or 16 blocks for local selections of the clipping points. The SSIM (Structural Similarity Index) [15] has been used to design the backlight power saving algorithm in [16] instead of the MSE. The work in [17] used the Gaussian distribution model to obtain the best clipping point for the LCD backlight module. For the analysis of different color spaces, the literature [18] takes RGB as the research object and automatically analyzes the color of urine test paper. A study on tooth color with HSV color space was proposed by [19]. Furthermore, a color space YEF transform was used to save the computational complexity. Based on human visual perception, the authors in [20] proposed a decomposition of the image intensity into an illumination layer and a reflectance layer, where the color saturation is also enhanced. A sub-band decomposition is proposed in [21] to preserve the luminance and contrast levels, while the excessive power consumption is prevented. In [22], a method of image fusion is proposed to detect the details of different images. For the common haze problem in dynamic images, a dehazing method with fast histogram fusion is proposed in [23] as a solution. They proposed a modified white balance algorithm to recognize and remove color veils (unbalanced color channels) to reduce the effect of unbalanced color channels ignored by an Atmospheric Illumination Prior (AIP). The renders balanced image contrast enhancement and inherent color preservation. In [24], an optimization problem is formed for video quality and power consumption. A modified shuffled frog leaping algorithm is proposed to solve the problem.

For the problem of the power consumption and the image quality, the above works are performed for considered frames. In [25], the disparity vector extrapolation technique was applied in the multi-view video transmission by using a frame loss concealment algorithm. Furthermore, the analysis of motion vectors in [26] provides a direction for thinking about this issue, focusing on the similarity of overlapping regions of adjacent frames, and the priority of target blocks in missing frames. In the proposed work, we consider the fact that in a video, the pixel characteristics of adjacent frames are similar, therefore the power-saving decisions (clipping points) should be sharable among consecutive frames with good performance. The most important benefit of this approach is saving the computation time of each frame. That is, if a certain criterion is met for the current frame, the power-saving algorithm will not be performed, and the power-saving decision (clipping point) for the current frame will automatically be the one decided in the previous frame, to save the execution time for power-saving method in the current frame. The criterion is designed based on the motion vector information that is already there during the decoding of the video image in the receiver at the display end, therefore there is no extra work for the information extraction. Furthermore, the algorithm is very simple to process the motion vector information so that the computation complexity of the whole proposed fast algorithm is low.

The novelties of the proposed method are as follows:To the best of our knowledge, the proposed method is the first work to use a motion vector to design the method for the backlight power-saving algorithm.The motion vector is used as the similarity measurement of the adjacent frames, and if qualified, the power-saving decision (the clipping point) of the current frame will automatically be the one of the previous frame, without activating the whole power-saving method in order to save the computation time.The availability of the motion vector is at no further expense, since it is the product accompanied with the decoding of the video images in the display/receiver end.The proposed algorithm of processing the motion vector is very simple to avoid the overhead of the proposed work.The combination of the above two points makes the proposed work a fast algorithm. The proposed work can be used with the existing methods with the use of clipping points.

The proposed method can eliminate 52.01%, 48.10%, 49.84%, and 64.22% of the processing time of the works I2GEC [13], MGEC4 [14], MGEC16 [14], and Gaussian [17], respectively, while the performances of image quality and power reduction are maintained well (the differences are only −0.02~1.91 dB and −0.001~0.008 W, respectively).

## 2. Materials and Methods

### 2.1. Existing Clipping-Point-Based Backlight Power-Saving Algorithms

In this section, several state-of-the-art backlight power-saving methods will be introduced I2GEC [13], MGEC4 [14], MGEC16 [14], and Gaussian [17]; the above works are based on clipping points. The execution time of above works is to be reduced by the proposed method.

#### 2.1.1. I2GEC: Integrity-Based Gray-Level Error Control

I2GEC (Integrity-based Gray-level Error Control) [13] uses the best-known image quality index called PSNR (Peak Signal-to-Noise Ratio) to find the clipping point with the lowest power consumption at the target quality. The formula for PSNR is as follows:(1)$$\mathrm{MSE}=\frac{1}{N}\overset{N}{\underset{i=1}{\sum }}{\left(x_{i}-y_{i}\right)}^{2}$$
(2)$$\mathrm{PSNR}=10log_{10}\left(\frac{255^{2}}{\mathrm{MSE}}\right)$$
where $x_{i}$ is the pixel in original image, $y_{i}\;$ is the pixel in processed image, *N* is the total number of pixels in the image. In this method, for a given target PSNR, the mean-square error (MSE) and the total mean square error ($TSE_{T}$) is calculated:(3)$$TSE_{T}=MSE_{T}\times \lambda \;,\;\lambda =\lambda_{row}\times \lambda_{col}\times \lambda_{color}.\;$$
(4)$$MSE_{T}=\frac{255^{2}}{10^{PSNR/10}\;}$$
where $\lambda_{row},\lambda_{col\;}$ are the length and width of the image. Since color images are composed of three primary colors (RGB), the $\lambda_{color}$ equals to 3. The algorithm sets the clipping point to make the processed image satisfying the target $TSE_{T}$. When the clipping point is set to be $I_{ccp}$, $0\leq I_{CCP}\leq I_{MAX}$, the resulting degradation of the image $TSE_{C}$ is computed by:(5)$$TSE_{R\_C}=\overset{I_{MAX}}{\underset{i=I_{ccp-1}}{\sum }}H_{R}\left(i\right)\times {\left(i-I_{ccp}\right)}^{2},.\;$$
(6)$$TSE_{G\_C}=\overset{I_{MAX}}{\underset{i=I_{ccp-1}}{\sum }}H_{G}\left(i\right)\times {\left(i-I_{ccp}\right)}^{2},$$
(7)$$TSE_{B\_C}=\overset{I_{MAX}}{\underset{i=I_{ccp-1}}{\sum }}H_{B}\left(i\right)\times {\left(i-I_{ccp}\right)}^{2},$$
(8)$$TSE_{C}=TSE_{R\_C}+TSE_{G\_C}+TSE_{B\_C}.\;$$
$H_{R}\left(i\right),\;H_{G},\;\mathrm{and}\;H_{B}\left(i\right)$ represents the number of *R*, *G*, and *B* pixels corresponding to gray level *i*. The clipping point $I_{ccp}$ is tested starting from $I_{MAX}$ to see if the $TSE_{C}$ satisfies $TSE_{T}$. If not, the clipping point $I_{ccp}$ is decreased by 1, and then the resulting $TSE_{C}$ will be checked again. This procedure will repeat until the condition is satisfied. As can be seen, the complexity of the method is high.

#### 2.1.2. MGEC: Multi-Histogram-Based Gray Level Error Control

MGEC (Multi-histogram-based Gray Level Error Control) [14] has a better performance on the PSNR and power-saving performances since it is a block-based algorithm. Assuming the image is divided into *M* × *N* blocks, the mean-square error (MSE) and the equivalent target $TSE_{T}$ is computed as:(9)$$MSE_{T}=\frac{255^{2}}{10^{PSNR/10}}\;$$
(10)$$TSE_{T}=MSE_{T}\times \frac{\lambda_{row}}{M}\times \frac{\lambda_{col}}{N}\times \lambda_{color}$$

For each clipping point $I_{ccp}$, the image quality degradation $TSE_{C}^{n}$ is computed for each block, *n* = 1~*M* × *N*:
(11)$${TSE}_{R\_C}^{n}=\overset{I_{MAX}}{\underset{i=I_{ccp}}{\sum }}H_{R}^{n}\left(i\right)\times {\left(i-I_{ccp}\right)}^{2}$$
(12)$${TSE}_{G\_C}^{n}=\overset{I_{MAX}}{\underset{i=I_{ccp}}{\sum }}H_{G}^{n}\left(i\right)\times {\left(i-I_{ccp}\right)}^{2}$$
(13)$${TSE}_{B\_C}^{n}=\overset{I_{MAX}}{\underset{i=I_{ccp}}{\sum }}H_{B}^{n}\left(i\right)\times {\left(i-I_{ccp}\right)}^{2}.\;$$
(14)$$TSE_{C}^{n}=TSE_{R\_C}^{n}+TSE_{G\_C}^{n}+TSE_{B\_C}^{n}.\;$$

$H_{R}\left(i\right),\;H_{G},\;\mathrm{and}\;H_{B}\left(i\right)$ represents the number of *R*, *G*, and *B* pixels corresponding to gray level *i*. After obtaining the $TSE_{C}^{n}$ for all blocks, the maximum $TSE_{C}^{n}$ is choosen as $TSE_{M}$:(15)$$TSE_{M}=\arg\underset{n}{\max}\left\{TSE_{C}^{n}\right\}$$

$TSE_{M}$ is then further compared with the target $TSE_{T}$. Similar to I2GEC, if the $TSE_{M}$ does not satisfy $TSE_{T}$, the clipping point $I_{ccp}$ is decreased by 1, and then the resulting $TSE_{M}$ is checked again. The procedure is repeated until the condition is satisfied. Again, the complexity of the method is high. This paper works with the four-segment version, denoted as MGEC4, and the sixteen-segment version, denoted as MGEC16.

#### 2.1.3. Gaussian: Adaptive Local Dimming Backlight Control Based on Gaussian Distribution

The state-of-the-art method, adaptive local dimming backlight control based on Gaussian distribution [17], is denoted as Gaussian. The backlight dimming algorithm is first performed with the same concept that the clipping point will be the largest pixel value where $\;x_{i}\;$ is the pixel in original image, N is the total number of pixels in the image. The Gaussian distribution of the image pixels is first formulated:(16)$$\sigma =\sqrt{\frac{1}{N}\overset{N}{\underset{i=1}{\sum }}{\left(x_{i}-\mu \right)}^{2}}$$
(17)$$f\left(x;\mu ,\sigma \right)=\frac{1}{\sigma \sqrt{2\pi }}e^{\frac{-{\left(x-\mu \right)}^{2}}{2\sigma^{2}}}$$

In order to improve the power-saving rate and maintain high image quality, and according to the normal distribution probability, the clipping point $Cp_{\left(m,n\right)}$ for each block is found where Z means the distance between the expected probability and the maternal average of luminance by
(18)$$Cp_{\left(m,n\right)}=Z\times \sigma +\mu e^{\left(\frac{-x^{2}}{2}\right)}$$

The pixel compensator and the PWN (power width modulation) module will then be applied for the algorithm.

#### 2.1.4. Power Model

The descriptions of the above algorithms focus on the aspects of the resulting image quality given a certain setting of the clipping point. The other aspect of the backlight power-saving algorithms is the power consumption for a given clipping point. All the above works (I2GEC [13], MGEC4 [14], MGEC16 [14] and Gaussian [17]) follow the same power model, which is derived in [27]:(19)$$\begin{array}{c} P_{backlight}\left(\beta \right)=\{\begin{array}{c} A_{lin}\cdot \beta +C_{lin}\;\;0\leq \beta \leq C_{s}\\ A_{sat}\cdot \beta +C_{sat}\;\;C_{s}<\beta \leq 1 \end{array}\;\\ \beta =\frac{clipping\;point}{255},\;A_{lin}=1.9600,\;A_{sat}=6.9440,\\ C_{lin}=-0.2372\;\;C_{sat}=-4.3240,\;C_{s}=0.8234 \end{array}\}$$

In general, lower clipping point produces lower PSNR and lower power consumption. Table 1 tabulates the power of some example clipping points.

### 2.2. Fast Algorithm for Power Control and Image-Quality Control Using Motion Vectors

In this section, we proposed to use the motion vectors decoded at the decoder as the important information to estimate the similarity among adjacent frames. An associated algorithm is designed to speed up state-of-the-art methods with very slight performance degradation.

#### 2.2.1. Motion Vector Estimation (Performed at the Encoder)

In the video encoder, the previous frame served as the reference frame to reduce the bit rate for storing the information of the current frame and to check if there are block pixels similar to the block pixels in the current frame. The displacement of the current block to the reference block describes the “motion” of the object, moving from previous (frame) to the current frame. The displacement is defined by a set of “motion vector”, (*mvx*, *mvy*). The process of finding the optimal motion vector in the encoder is called “motion estimation” [28].

The motion estimation is proceeded as follows. The prediction error, defined as *D*, of a block refers to the pixel differences between the current block and the reference block in the reference frame. The number of bits to store the associated motion vector (*mvx*, *mvy*) is recorded as *R*. To find the best motion vector (*mvx*, *mvy*), one of the most basic methods is to perform search within a fixed range in the reference frame, finding the testing block in the reference frame with the minimum (optimal) *L* = *D* + λ*R*, to have minimal weighted sum of prediction error and bitrate. The search illustration can be seen in Figure 1. The best (optimal) motion vector is then recorded in the video bit stream to be stored in the hard drive or transmitted over the network.

As can be seen, the motion estimation is a complex process. However, we have to note that this complex process is performed in the encoder, as opposed to the decoder at the display side, where the proposed backlight saving algorithm is performed. Therefore, the complexity of the motion estimation has nothing to do with the complexity of the proposed backlight saving algorithm.

#### 2.2.2. The Proposed Power-Saving Method Using Motion Vectors

The correlation between the current frame and the reference frame (previous frame) can be characterized by the motion vector in some sense. The differences between the current frame and the previous frame are small if the values of motion vectors are small. In this case, the correlation between the current frame and the reference frame (previous frame) is very high. On the other hand, if the motion vectors are so large that the differences between the current frame and the previous frame are very large, the correlation between the current frame and the reference frame (previous frame) is very small. This is the design concept of the proposed power-saving method.

What is more, there is no extra effort required to obtain the motion vector in the display side. This is because, in order to decode the videos at the receiver, the motion vectors in the bitstream are first decoded to reconstruct the frame [28]. All we have to do is to store them for the proposed fast backlight-saving algorithm after the reconstruction of the video, instead of throwing them away. Therefore, the availability of the motion vectors for each frame for our application is at no cost, and thus will not increase the complexity of our algorithm.

The proposed algorithm is designed as follows: the regular power-saving method is applied for the first frame of the video. For the following frames, the motion vectors are used to analyze the correlation between the current frame and the previous frame. For each pair of the motion vector (*mvx_i_*, *mvy_i_*) of a block *i*, the magnitude of the motion vector is computed as
(20)$${\mathrm{Mag}}_{i}=\left|mvx_{i}\right|+\left|mvy_{i}\right|$$

The Mag*_i_* for all the blocks in a frame are summed to be SumMag:(21)$$\mathrm{SumMag}=\underset{i}{\sum }{\mathrm{Mag}}_{i\;\;\;\;}$$

If SumMag is smaller than a threshold, it shows that the current image and the previous image have small difference, therefore, the proposed algorithm does not need to perform the whole regular method again. Instead, the proposed algorithm can automatically use the clipping point decision of the previous frame (reference frame).

By doing this, the proposed algorithm saves the computational complexity of finding the clipping point for the current frame. If SumMag is greater than a threshold, it means that the current frame and the previous frame have the huge difference. Therefore, the proposed algorithm needs to perform the whole regular method for the current frame. The overall algorithm is illustrated in Figure 2.

It is obvious that when the threshold is set to be small, it means that it is easy to be over the threshold to re-run the whole regular method; this does not result in time saving, but the image quality is maintained. On the other hand, if the threshold is set to be large, it means that it is easy to be under the threshold, so that the proposed method automatically takes the clipping point of the previous frame frequently; this results in large time saving, but with possible drops in image quality. In the experiment section, we set the various threshold to reveal the tradeoffs.

## 3. Results

In this section, the experimental results are presented. As discussed, the proposed work aims to reduce the execution time of the existing works while keeping the performances by using the information of the decoded motion vectors. The proposed method works as the flow chart shown in Figure 2. The existing works focus on the methods discussed earlier, I2GEC [13], MGEC4 [14], MGEC16 [14], and Gaussian [17], as the regular power-saving algorithm in Figure 2. The test videos are *NASASOF-WindTunnelTesting_100*, *NASASF-ISSLife_0324*, *NASASOF-WindTunnelTesting_0324*, *NASA-EPAQ_0325* and *NASASF-FOT_0325*.

The proposed method works as the flow in Figure 2. The thresholds set in Figure 2 are 0, 15,000, and 30,000 to reveal the various performances under different settings. Note that the setting of threshold equal to 0 corresponds to the original/regular power-saving algorithm, since it is always executed for every frame. In Appendix A, several performances are evaluated: execution time (seconds) in Table A1, PSNR (Peak Signal-to-Noise Ratio, dB) in Table A2, power (watts) in Table A3, and selected clipping points in Table A4. The average performances over different videos for a specific method and threshold are shown in Figure 3, Figure 4, Figure 5 and Figure 6, respectively. Note that all the methods are realized in software and run on a personal computer with CPU i7-7700K, and the measurement of time (in seconds) is the interval of the beginning of the method and the end of the method for all video frames.

The average time comparisons in Table A1 are first discussed. As can be seen, for I2GEC, the original work (threshold = 0) takes 28.88 s, and when the proposed work participates with the threshold being 15,000, the time reduced to 21.69 secs and the DT (decreased time, %) is 25.21%. When the threshold is set with the even higher value of 30,000, the time is further reduced to 13.86 secs, with DT = 52.01%. For MGEC4, the threshold 15,000 has DT 26.48%, and the threshold 30,000 has 48.10%. For MGEC16, the threshold 15,000 has DT 28.28%, and the threshold 30,000 has 49.84%. For Gaussian, the threshold 15,000 has DT 37.78%, and the threshold 30,000 has 64.22%. Figure 3 provides bar comparisons for the above descriptions. As can be observed, a higher threshold induces higher DT since the higher threshold allows higher chances to simply use the clipping point from the previous frame. Therefore, the proposed work can indeed reduce the execution time of the existing methods, and the reduction can vary with different settings of the threshold.

For the comprehensive comparisons, the average PSNR (Table A2, Figure 4), power consumption (Table A3, Figure 5; computed by Equation (19)) and the selected clipping points (Table A4, Figure 6) are discussed together. As discussed, the proposed work can help I2GEC with DT = 25.21% and 52.01%. What is more, the PSNR differences are only 0.01 dB, the power difference is only 0.006 W and 0.007 W, and the clipping points are the same on average. These indicate that the proposed work can not only reduce the execution time by good ratios, but also maintain the quality of the work. For MGEC4, the DTs in the proposed work are 26.48% and 48.10%. The performances are also maintained since the PSNR differences are only −0.02 dB and 1.91 dB, power differences are only 0.006 W and 0.007 W, and the clipping points are the same on average. Similarly, for MGEC16, the proposed work can provide DT = 28.28% and 49.84%. These good reductions in time do not affect the performances where the PSNR differences are only 0.16 dB and 0.19 dB, the power differences are only 0.007 W and 0.008 W, and the selected clipping points are the same again. Finally, for Gaussian, when the DTs caused by the proposed work are 37.78% and 64.22%, the performances are not degraded as the PSNR differences are 0.00 dB and 0.01 dB, the power differences are −0.001 W and 0.000 W, and the selected clipping points are the same on average.

We have to note that the proposed work does not aim to improve or degrade the PSNR and the power performance of the existing works; the proposed work aims to position the PSNR and the power performance close to those of the existing works. Therefore, it does not matter if the differences in the above-mentioned tables are positive or negative; we focus on the magnitudes of the differences being small.

## 4. Conclusions

This paper presents an innovative fast algorithm to reduce the computation time of the existing display backlight control energy-saving methods. The proposed work is based on the analysis of the motion vectors decoded for each video frame before the display. The motion vectors were used as the similarity measurement of the adjacent frames. If the adjacent frames are similar, the selected clipping point, which is the backlight algorithm decision of the current frame can simply duplicate that of the reference frame, without needing to be computed by the original algorithm; this reduces the execution time of the current frame for the clipping point by the backlight-saving algorithm. Working with several state-of-the-art methods, under different similarity threshold settings, the execution time was reduced by 25.21~64.22%. In addition, the proposed work does not make significant differences in PSNR performances (−0.02~1.91 dB) and power consumption (−0.001~0.008 W) with those of the state-of-the-art methods.

## Figures and Tables

**Figure 1 sensors-22-07170-f001:**
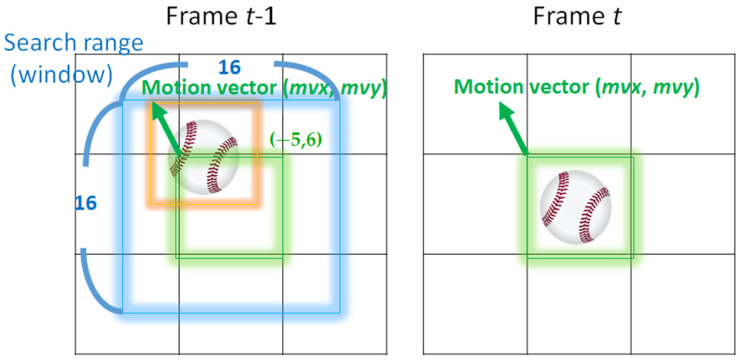
The illustration of motion estimation and motion vector (mvx, mvy).

**Figure 2 sensors-22-07170-f002:**
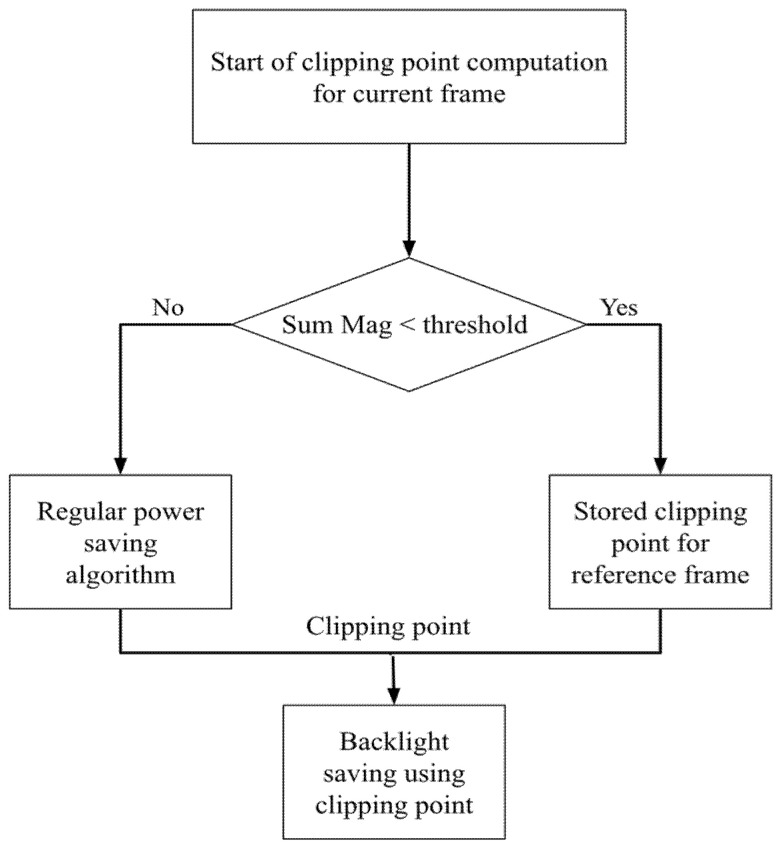
Flow chart of the proposed work.

**Figure 3 sensors-22-07170-f003:**
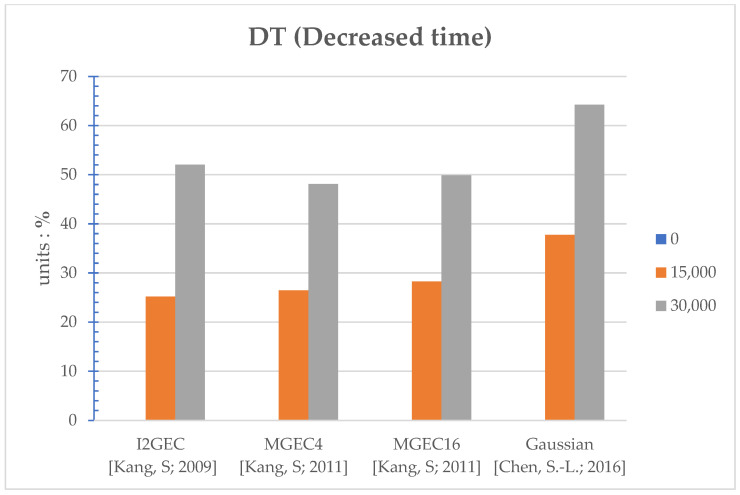
Bar comparisons of the DT (decreased time, %) of different algorithms at different thresholds [9,10,13].

**Figure 4 sensors-22-07170-f004:**
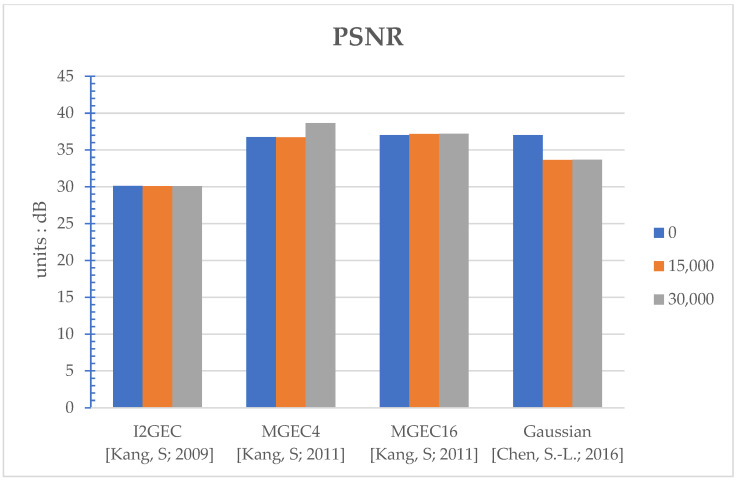
Bar comparisons of PSNR (dB) of different algorithms at different thresholds [9,10,13].

**Figure 5 sensors-22-07170-f005:**
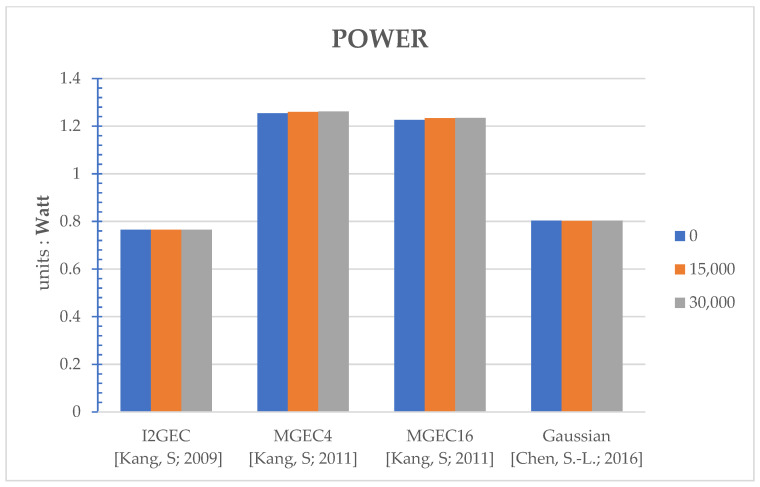
Bar comparisons of power (watts) of different algorithms at different thresholds [9,10,13].

**Figure 6 sensors-22-07170-f006:**
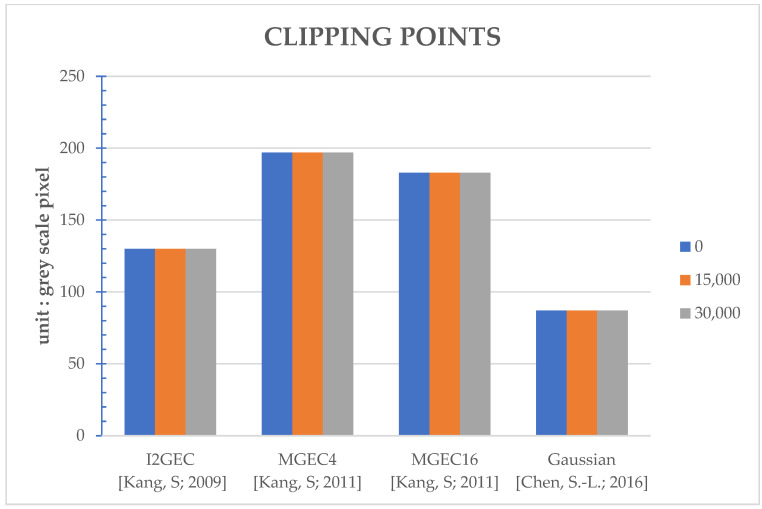
Bar comparisons of the selected clipping points (in unit of gray scale pixel) of different algorithms at different thresholds [9,10,13].

**Table 1 sensors-22-07170-t001:** The example of the clipping points and the corresponding power consumption (computed by Equation (19)).

Clipping Points	Power (Watt)
255	2.6200
200	1.3001
150	0.9157
100	0.5314
50	0.1471

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
