# Peer review of "Fast Control for Backlight Power-Saving Algorithm Using Motion Vectors from the Decoded Video Stream"

_sensors, 2022, doi:10.3390/s22197170_

Round 1

Reviewer 1 Report

This paper introduces the motion vector in video coding into the backlight power algorithm. The motion vector judges the similarity between frames, and the decision of the previous frame is directly adopted for similar frames, thus avoiding repeated operations and reducing the operation complexity. This method is applied to I2GEC, MGEC4, MGEC16, and Gaussian algorithms. The data show that the execution time of these algorithms is reduced by 25.21%~64.22% without a noticeable difference in power and image quality.

Before Sensors could accept the manuscript, I have two concerns.

1. SumMag is calculated by Eq. (21) as a similarity index. What is the block number of motion vectors for each algorithm? Is it the same? How will the motion vectors with different block numbers affect the algorithm results?

2. The display information in Table 5 and Figure 6 is somewhat repetitive. Thresholds should not influence the selection of the clipping points for the same algorithm. Secondly, the clipping point should be different for each frame of the video, so is the data in Table 5 the average of clipping points of all frames of the video?

And some format Issues:

1. In Figure 2, the flow chart occupies too much space.

2. The clarity of Figures 3-6 is inconsistent, and the fonts in the figures are inconsistent.

Reviewer 2 Report

A simple and effective clipping point searching algorithm is proposed to control the power of backlight in LCD. The method uses the motion vector information as a reference to determine if the clipping point should be updated. In this way, the new method shows less execution time, similar image quality and power consumption. I recommend this manuscript for publishing after major revisions. Some specific questions are listed below.

11. In section 2, several equations are not clear to me. For example, in Eq. 5, what does capital letter H mean? It appears several times but there is no explaining.

22.  Data in table 2-5 seems too much to readers. Please consider put it in the supplementary material. Figure 3-6 should be enough to tell the story.

Reviewer 3 Report

 This paper presents an innovative fast algorithm to reduce the computation time of the existing display backlight control energy-saving methods.

A few suggestions for the author's reference:

1. Figures 3,4,5, and 6  Y-axis should be marked with units to avoid confusing readers.

2. The resolution of Figures 4-6 needs to be improved.

The mainstream of the current market backlight includes MiniLEDs backlight technology, due to the consideration of commercial competitiveness, the number of light sources is much smaller than the number of pixels.

3. Common forms of backlight include edge light and direct type. Please explain whether the calculation method proposed in this article is suitable for edge type or direct type backlight, or is it universal?

4. Please explain whether the algorithm proposed in this article has limitations in the batch allocation between light sources and pixels and how to achieve the best allocation (for example, how many light sources are in a dimming zone).

5. Are different LCD aspect ratio calculation methods common or need to be adjusted?

6. In the introduction part, 3 years of research related to dynamic backlight control of mini LEDs can be added, which also needs to explain the differences with this article.

The above suggestions, if the author can further describe the explanation, I suggest accepting this manuscript.
